# OUTLIER-ROBUST GROUP INFERENCE VIA GRADIENT SPACE CLUSTERING

## ABSTRACT

Traditional machine learning models focus on achieving good performance on the overall training distribution, but they often underperform on minority groups. Existing methods can improve the worst-group performance, but they can have several limitations: (i) they require group annotations, which are often expensive and sometimes infeasible to obtain, and/or (ii) they are sensitive to outliers. Most related works fail to solve these two issues simultaneously as they focus on conflicting perspectives of minority groups and outliers. We address the problem of learning group annotations in the presence of outliers by clustering the data in the space of gradients of the model parameters. We show that data in the gradient space has a simpler structure while preserving information about minority groups and outliers, making it suitable for standard clustering methods like DBSCAN. Extensive experiments demonstrate that our method significantly outperforms state-of-the-art both in terms of group identification and downstream worst-group performance.

## 1 INTRODUCTION

Empirical Risk Minimization (ERM), i.e., the minimization of average training loss over the set of model parameters, is the standard training procedure in machine learning. It yields models with strong in-distribution performance[1] but does not guarantee satisfactory performance on minority groups that contribute relatively few data points to the training loss function (Sagawa et al., 2019; Koh et al., 2021). This effect is particularly problematic when the minority groups correspond to socially-protected groups. For example, in the toxic text classification task, certain identities are overwhelmingly abused in online conversations that form data for training models detecting toxicity (Dixon et al., 2018). Such data lacks sufficient non-toxic examples mentioning these identities, yielding problematic and unfair spurious correlations – as a result ERM learns to associate these identities with toxicity (Dixon et al., 2018; Garg et al., 2019; Yurochkin & Sun, 2020). A related phenomenon is *subpopulation shift* (Koh et al., 2021), i.e., when the test distribution differs from the train distribution in terms of group proportions. Under subpopulation shift, poor performance on the minority groups in the train data translates into poor overall test distribution performance, where these groups are more prevalent or more heavily weighted. Subpopulation shift occurs in many application domains (Tatman, 2017; Beery et al., 2018; Oakden-Rayner et al., 2020; Santurkar et al., 2020; Koh et al., 2021).

Prior work offers a variety of methods for training models robust to subpopulation shift and spurious correlations, including group distributionally robust optimization (gDRO) (Hu et al., 2018; Sagawa et al., 2019), importance weighting (Shimodaira, 2000; Byrd & Lipton, 2019), subsampling (Sagawa et al., 2020; Idrissi et al., 2022; Maity et al., 2022), and variations of tilted ERM (Li et al., 2020; 2021). These methods are successful in achieving comparable performance across groups in the data, but they require group annotations. The annotations can be expensive to obtain, e.g., labeling spurious backgrounds in image recognition (Beery et al., 2018) or labeling identity mentions in the toxicity example. It also could be challenging to anticipate all potential spurious correlations in advance, e.g., it could be background, time of day, camera angle, or unanticipated identities subject to harassment.

Recently, methods have emerged for learning group annotations (Sohoni et al., 2020; Liu et al., 2021; Creager et al., 2021) and variations of DRO that do not require groups (Hashimoto et al., 2018; Zhai

---

[1]I.e. low loss on test data drawn from the same distribution as the training dataset.

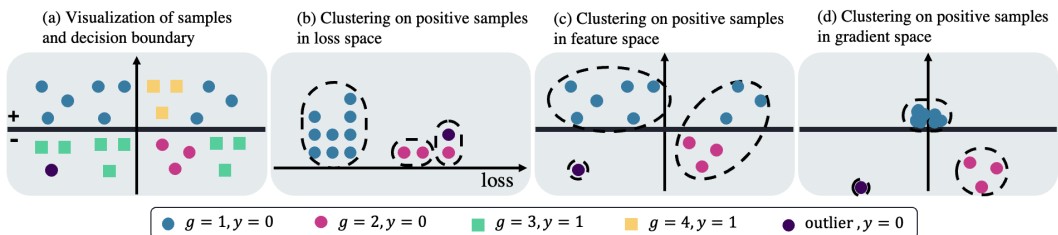

Figure 1: **An illustration of learning group annotations in the presence of outliers.** (a) A toy dataset in two dimensions. There are four groups $g = 1, 2, 3, 4$ and an outlier. $g = 1$ and $g = 3$ are the majority groups distributed as mixtures of three components each; $g = 2$ and $g = 4$ are unimodal minority groups. $y$-axis is the decision boundary of a logistic regression classifier. Figures (b, c, d) compare different data views for learning group annotations and detecting outliers via clustering of samples with $y = 0$. (b) loss values can confuse outliers and minority samples which both can have high loss; (c) in the original feature space it is difficult to distinguish one of the majority group modes and the minority group; (d) gradient space (bias gradient omitted for visualization) simplifies the data structure making it easier to identify the minority group and to detect outliers.

et al., 2021). One common theme is to treat data where an ERM model makes mistakes (i.e., high-loss points) as a minority group (Hashimoto et al., 2018; Liu et al., 2021) and increase the weighting of these points. Unfortunately, such methods are at risk of overfitting to outliers (e.g., mislabeled data, corrupted images), which are also high-loss points. Indeed, existing methods for outlier-robust training propose to *ignore* the high-loss points (Shen & Sanghavi, 2019), the direct opposite of the approach in (Hashimoto et al., 2018; Liu et al., 2021).

In this paper, our goal is to learn group annotations in the presence of outliers. Rather than using loss values (which above were seen to create opposing tradeoffs), we propose to instead first represent data using gradients of a datum's loss w.r.t. the model parameters. Such gradients tell us how a specific data point wants the parameters of the model to change to fit it better. In this gradient space, we anticipate groups (conditioned on label) to correspond to gradients forming clusters. Outliers, on the other hand, majorly correspond to isolated gradients: they are likely to want model parameters to change differently from any of the groups *and* other outliers. See Figure 1 for an illustration. The gradient space structure allows us to separate out the outliers and learn the group annotations via traditional clustering techniques such as DBSCAN (Ester et al., 1996). We use learned group annotations to train models with improved worst-group performance (measured w.r.t. the true group annotations).

We summarize our contributions below:

- We show that gradient space simplifies the data structure and makes it easier to learn group annotations via clustering.
- We propose Gradient Space Partitioning (GRASP), a method for learning group annotations in the presence of outliers for training models robust to subpopulation shift.
- We conduct extensive experiments on one synthetic dataset and three datasets from different modalities and demonstrate that our method achieves state-of-the-art both in terms of group identification quality and downstream worst-group performance.

## 2 PRELIMINARIES AND RELATED WORK

In this section, we review the problem of training models in the presence of minority groups. Denote $[N] = \{1, \ldots, N\}$. Consider a dataset $\mathcal{D} = \{\mathbf{z}\}_{i=1}^{n} \subset \mathcal{Z}$ consisting of $n$ samples $\mathbf{z} \in \mathcal{Z}$, $\mathbf{z} = (\mathbf{x}, \mathbf{y})$, where $\mathbf{x} \in \mathcal{X} = \mathbb{R}^d$ is the input feature and $\mathbf{y} \in \mathcal{Y} = \{1, \ldots, C\}$ is the class label. The samples from each class $y \in \mathcal{Y}$ are categorized into $K_y$ groups. Denote $K$ to be the total number of groups $\{\mathcal{G}_1, \ldots, \mathcal{G}_K\} \triangleq P \subset \mathcal{Z}$, where $K = \sum_{y \in \mathcal{Y}} K_y$ and $\mathcal{G}_k \bigcap \mathcal{G}_{k'} = \emptyset$ for all pairs $k \neq k' \in [K]$. Denote the group membership of each point in the dataset as $\{\mathbf{g}_i\}_{i=1}^{n}$, where $\mathbf{g}_i \in [K]$ for all $i \in [n]$. For example, in toxicity classification, a group could correspond to a toxic comment mentioning

a specific identity, or, in image recognition, a group could be an animal species appearing on an atypical background (Beery et al., 2018; Sagawa et al., 2019).

The goal of learning in the presence of minority groups is to learn a model $h \in \mathcal{H} : \mathcal{X} \to \mathcal{Y}$ parameterized by $\boldsymbol{\theta} \in \Theta$ that performs well on all groups $\mathcal{G}_k$, where $k \in [K]$. Depending on the application, this model can alleviate fairness concerns (Dixon et al., 2018), remedy spurious correlations in the data (Sagawa et al., 2019), and promote robustness to subpopulation shift (Koh et al., 2021), i.e., when the test data has unknown group proportions.

We divide the approaches for learning in the presence of minority groups into three categories: the *group-aware* setting where the group annotations $\mathrm{g}_i$ are known, the *group-oblivious* setting that does not use the group annotations, and the *group-learning* setting where the group annotations are learned from data to be used as inputs to the group-aware methods.

**Group-aware setting.** Many prior works assume access to the minority group annotations. Among the state-of-the-art methods in this setting is group Distributionally Robust Optimization (gDRO) (Sagawa et al., 2019). Let $\ell : \mathcal{Y} \times \mathcal{Y}$ be a loss function. The optimization problem of gDRO is

$$\min_{\boldsymbol{\theta} \in \Theta} \max_{k \in [K]} \frac{1}{|\mathcal{G}_k|} \sum_{\mathbf{z} \in \mathcal{G}_k} \ell(\mathrm{y}, h_{\boldsymbol{\theta}}(\mathbf{x})), \qquad \text{(gDRO)}$$

which aims to minimize the maximum group loss. In addition to assuming clean group annotations, another line of research under this setting considers noisy or partially available group annotations (Jung et al., 2022; Lamy et al., 2019; Mozannar et al., 2020; Celis et al., 2021). Methods in this class achieve meaningful improvements over ERM in terms of worst-group accuracy, but anticipating relevant minority groups and obtaining the annotations is often burdensome.

**Group-oblivious setting.** In contrast to the group-aware setting, the *group-oblivious* setting attempts to improve worst-group performance without group annotations. Methods in this group rely on various forms of DRO (Hashimoto et al., 2018; Zhai et al., 2021) or adversarial reweighing (Lahoti et al., 2020). Algorithmically, this results in up/down-weighing the contribution of the high/low-loss points. For example, Hashimoto et al. (2018) optimizes a DRO objective with respect to a chi-square divergence ball around the data distribution, which is equivalent to minimizing $\frac{1}{n} \sum_i [\ell(\mathrm{y}, h_{\boldsymbol{\theta}}(\mathbf{x})) - \eta]_+^2$, i.e., an ERM discounting low-loss points by a constant depending on the ball radius.

**Group-learning setting.** The final category corresponds to a two-step procedure, wherein the data points are first assigned group annotations based on various criteria, followed by group-aware training typically using gDRO. In this category, Just Train Twice (JTT) (Liu et al., 2021) trains an ERM model and designates high-loss points as the minority and low-loss points as the majority group; George (Sohoni et al., 2020) seeks to cluster the data to identify groups with a combination of dimensionality reduction, overclustering, and augmenting features with loss values, and Environment Inference for Invariant Learning (EIIL) (Creager et al., 2021) finds group partition that maximizes the Invariant Risk criterion (Arjovsky et al., 2019).

Our method, Gradient Space Partitioning (GraSP), belongs to this category. We provide a mind map of our problem setting in Fig. 3 for clearer explanation. GraSP differs from prior works in its ability to account for outliers in the data. In addition, prior methods in this and the group-oblivious categories typically require validation data with *true* group annotations for model selection to achieve meaningful worst-group performance improvements over ERM, while GraSP does not need these annotations to achieve good performance. In our experiments, this can be attributed to GraSP's better recovery of the true group annotations, making them suitable for gDRO model selection (see Section 4). We summarize properties of the most relevant methods in each setting in Table 1.

**The challenge of outliers.** Outliers, e.g., mislabeled samples or corrupted images, are ubiquitous in applications (Singh & Upadhyaya, 2012), and outlier detection has long been a topic of inquiry in ML (Hodge & Austin, 2004; Wang et al., 2019). Outliers are especially challenging to detect when data has (unknown) minority groups, which could be hard to distinguish from outliers but require the opposite treatment: minority groups need to be upweighted while outliers must be discarded. Hashimoto et al. (2018) writes, "it is an open question whether it is possible to design algorithms which are both fair to unknown latent groups and robust [to outliers]."

We provide an illustration of a dataset with minority groups and an outlier in Figure 1(a). Figure 1(b) illustrates the problem with the methods relying on the loss values. Specifically, Liu et al. (2021) and

Table 1: **Summary of methods for learning in the presence of minority groups.** "-" indicates that there is no clear evidence in the prior works.

| Setting | | | Group-aware | Group-oblivious | | Group-learning | | | |
|---|---|---|---|---|---|---|---|---|---|
| Method | ERM | | gDRO (Sagawa et al., 2019) | $\chi^2$-DRO (Hashimoto et al., 2018) | DORO (Zhai et al., 2021) | JTT (Liu et al., 2021) | EIIL (Creager et al., 2021) | George (Sohoni et al., 2020) | GRASP (Ours) |
| Improves worst-group performance? | ✗ | | ✓ | ✓ | ✓ | ✓ | ✓ | ✓ | ✓ |
| No training group annotations? | ✓ | | ✗ | ✓ | ✓ | ✓ | ✓ | ✓ | ✓ |
| No validation group annotations? | ✓ | | ✗ | ✗ | ✗ | ✗ | ✗ | ✓ | ✓ |
| Group inference? | ✗ | | ✗ | ✗ | ✗ | ✓ | ✓ | ✓ | ✓ |
| Robust to outliers? | ✗ | | - | ✗ | ✗ | ✗ | - | - | ✓ |

Hashimoto et al. (2018) upweigh high-loss points, overfitting the outlier. Zhai et al. (2021) optimize Hashimoto et al. (2018)'s objective function after discarding a fraction of points with the largest loss values to account for outliers. They assume that outliers will have higher loss values than the minority group samples, which can easily be violated leading to exclusion of the minority samples, as illustrated in Figure 1.

**Gradients as data representations.** Given a model $h_{\boldsymbol{\theta}_0}(\cdot)$ and loss function $\ell(\cdot, \cdot)$, one can consider an alternative representation of the data where each sample is mapped to the gradient with respect to the model parameters of the loss on this sample:

$$\boldsymbol{f}_i = \frac{\partial \ell(\mathbf{y}_i, h_{\boldsymbol{\theta}}(\mathbf{x}_i))}{\partial \boldsymbol{\theta}}\bigg|_{\boldsymbol{\theta}=\boldsymbol{\theta}_0} \quad \text{for } i = 1, \ldots, n. \tag{1}$$

We refer to (1) as the *gradient representation*. For scalability and efficiency, one can consider a subset of the model parameters for large models with a high number of parameters such as ResNet-50 He et al. (2016). Prior works considered gradient representations (Mirzasoleiman et al., 2020), as well as loss values (Shen & Sanghavi, 2019), for outlier-robust learning. Gradient representations have also found success in novelty detection (Kwon et al., 2020b), anomaly detection (Kwon et al., 2020a), and out-of-distribution inputs detection (Huang et al., 2021).

In this work, we show that, unlike loss values, gradient representations are suitable for simultaneously learning group annotations *and* detecting outliers. Compared to the original feature space, gradient space simplifies the data structure, making it easier to identify minority groups. Figure 1(c) illustrates a failure of feature space clustering. Here the majority group for class $y = 0$ is a mixture of three components with one of the components being close to the minority group in the feature space. In the gradient space, for a logistic regression model, representations of misclassified points remain similar to the original features, while the representations of correctly classified points are pushed towards zero. We illustrate the benefits of the gradient representations in Figure 1(d) and provide additional details in the subsequent section.

## 3 GRASP: GRADIENT SPACE PARTITIONING

In this section, we present our method for group inference and outlier detection, which we refer to as Gradient Space Partitioning (GRASP). We first demonstrate that the gradient space is more suitable for using clustering methods to learn group annotations and identify outliers than the feature space. We support this claim with an example using a logistic regression model and an empirical study of synthetic and semi-synthetic datasets. We then present the details of GRASP in Sec. 3.2.

### 3.1 GRADIENT SPACE VS FEATURE SPACE

**Logistic regression example.** We present an example based on the logistic regression model to better understand how using the gradient space simplifies the data structure and aids clustering.

Consider a binary classification problem ($\mathbf{y} \in \{0, 1\}$) and logistic regression model $\mathbb{P}(\mathbf{y} = 1|\mathbf{x}) = \sigma(\boldsymbol{w}^\top \mathbf{x} + b)$ trained on the given dataset $\mathcal{D}$, where $\sigma(\cdot)$ denotes the sigmoid function, $\boldsymbol{w}$ are the coefficients and $b$ is the bias. Recall that the logistic regression loss is

$$\ell(\mathbf{y}, \sigma(\boldsymbol{w}^\top \mathbf{x} + b)) = -\mathbf{y} \log(\sigma(\boldsymbol{w}^\top \mathbf{x} + b)) - (1 - \mathbf{y}) \log(1 - \sigma(\boldsymbol{w}^\top \mathbf{x} + b)).$$

---

**Algorithm 1:** GRASP

---

**Input** : DBSCAN hyperparameters $\epsilon$ and $m$

Train the ERM classifier $\theta_0 \leftarrow \arg\min_{\theta \in \Theta'} \sum_{\mathbf{z} \in \mathcal{D}} \ell(\mathrm{y}, h_\theta(\boldsymbol{x}))$ ;

**for** $\mathbf{z} \in \mathcal{D}$ **do**
    Compute its gradient $\boldsymbol{f} \leftarrow \frac{\partial \ell(\mathrm{y}, h_\theta(\mathbf{x}))}{\partial \theta} \big|_{\theta = \theta_0}$;

**for** $y \in \mathcal{Y}$ **do**
    Consider all samples $\{\mathbf{z}_i\} \subset \mathcal{D}$ with $\mathrm{y} = y$ and their corresponding gradients $\{\boldsymbol{f}_i\}$;
    $\mu_f \leftarrow \mathrm{mean}(\{\boldsymbol{f}_i\})$, compute the distance matrix $D$, where $D_{ij} = 1 - \frac{\langle \boldsymbol{f}_i - \mu_f, \boldsymbol{f}_j - \mu_f \rangle}{\|\boldsymbol{f}_i - \mu_f\| \cdot \|\boldsymbol{f}_j - \mu_f\|}$;
    Assign group annotations and identify outliers by performing DBSCAN clustering in
    gradient space: $\{\hat{\mathrm{g}}_i\} \leftarrow \mathrm{DBSCAN}(D, \epsilon, m)$, where $\hat{\mathrm{g}}_i = -1$ indicates outliers;

**Output :** Dataset with predicted group annotations $\mathcal{D}' \leftarrow \{(\mathbf{x}, \hat{\mathrm{g}}, \mathrm{y})\}_{\{\hat{\mathrm{g}} \neq -1, \mathbf{z} \in \mathcal{D}\}}$, where the
    detected outliers are removed

---

The gradient of this loss at point $\mathbf{z}$ w.r.t. $(\boldsymbol{w}, b)$ is

$$\boldsymbol{f} =: \nabla_{[\boldsymbol{w},b]} \ell(\mathrm{y}, \boldsymbol{w}^\top \mathbf{x} + b) = (\sigma(\boldsymbol{w}^\top \mathbf{x} + b) - \mathrm{y}) \begin{bmatrix} \mathbf{x} \\ 1 \end{bmatrix}. \tag{2}$$

Note that this gradient is simply a scaling of the data vector $\mathbf{x}$ by the error $(\sigma(\boldsymbol{w}^\top \mathbf{x} + b) - \mathrm{y}) \in [-1, 1]$, padded by an additional element (the bias entry) consisting of the error alone. In particular, note that when $\mathbf{z}$ is correctly classified, the scaling $(\sigma(\boldsymbol{w}^\top \mathbf{x} + b) - \mathrm{y})$ will be close to zero, and when it is incorrectly classified, the magnitude of the scaling will approach 1.

We interpret this gradient ((2)) through the lens of Euclidean distance ($\|\boldsymbol{f}_i - \boldsymbol{f}_j\|_2$) and centered cosine distance ($1 - \frac{\langle \boldsymbol{f}_i - \mu_f, \boldsymbol{f}_j - \mu_f \rangle}{\|\boldsymbol{f}_i - \mu_f\|_2 \cdot \|\boldsymbol{f}_j - \mu_f\|_2}$) metrics,[2] respectively. Recall that we apply clustering to each class independently.

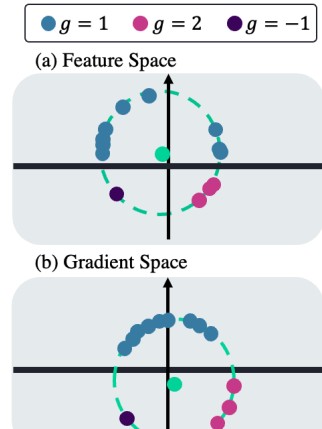

(a) Feature Space

(b) Gradient Space

- **Euclidean distance.** The scaling effect mentioned in the previous paragraph shrinks the correctly classified points towards the origin, while leaving the misclassified points almost unaffected. The error itself is included as an extra element (using loss as an additional feature was previously considered as a heuristic in feature clustering for learning group annotations (Sohoni et al., 2020)). Consequently, gradient clustering w.r.t. Euclidean distance should cluster the correctly classified samples into one "majority" group, and then divide the remaining points into minority groups and outliers based on the size of the error and their position in the feature space. For a visual example, see Figure 1(d).
- **Centered cosine distance.** We compare the (class conditioned; class dependency omitted for simplicity) centering terms in the gradient and feature spaces, $\mu_f$ and $\mu_x$ respectively:

$$\mu_f = \frac{1}{n} \sum_{i:y_i=c} (\sigma(\boldsymbol{w}^\top \mathbf{x}_i + b) - \mathrm{y}_i) \begin{bmatrix} \mathbf{x}_i \\ 1 \end{bmatrix}, \quad \mu_x = \frac{1}{n} \sum_{i:y_i=c} \mathbf{x}_i.$$

Figure 2: **Normalized representations of the data from Figure 1(a) in (a) feature space and (b) gradient space.** The green points are the means (before normalization) of the corresponding representations. Gradient space makes it easier to identify groups and detect outliers via clustering with centered cosine distances.

Due to the underrepresentation of the minority group in the data, the feature space center will be heavily biased towards the majority group which could hinder the clustering as illustrated in Figure 2(a). On the other hand, the expression of $\mu_f$ above implies that gradient space center upweighs high-loss points which are more representative of the minority groups, resulting in a center in-between minority and majority groups. Thus, centering in the gradient space facilitates learning group annotations via clustering with the cosine distance as illustrated in Figure 2(b).

---

[2]Here $\mu_f$ refers to the class-conditional empirical mean of $\boldsymbol{f}$.

Table 2: **Group identification quality of clustering methods in feature space and gradient space measured by Adjusted Rand Index (ARI).** Higher ARI indicated higher group identification quality. The results are reported on clean and contaminated versions of Synthetic and Waterbirds datasets. Three different clustering methods are considered: K-means, DBSCAN w.r.t. Euclidean distance (DBSCAN/Euclidean), and DBSCAN w.r.t. centered cosine distance (DBSCAN/Cos). We set $k = 2$ for K-means, which is the number of groups per class in these datasets. The gradient space clustering noticeably outperforms feature space clustering.

| DATASET | OUTLIERS? | | FEATURE SPACE | | | GRADIENT SPACE | |
|---------|-----------|---------|------------------|------------|---------|------------------|------------|
| | | K-MEANS | DBSCAN/EUCLIDEAN | DBSCAN/COS | K-MEANS | DBSCAN/EUCLIDEAN | DBSCAN/COS |
| SYNTHETIC | ✗ | .5505 | .5923 | .5133 | **.8409** | .7724 | .6943 |
| | ✓ | .3631 | .6042 | .4946 | .6436 | **.7237** | .6944 |
| WATERBIRDS | ✗ | .3932 | .0000 | .0418 | .7235 | .7304 | **.7453** |
| | ✓ | .3932 | .0000 | .0418 | .7171 | .7304 | **.7453** |

**Quantitative comparison.** We compare the group identification quality of clustering in feature space and gradient space on two datasets consisting of 4 groups. We consider both clean and contaminated versions. The first dataset is Synthetic based on the Figure 1 illustration. The second dataset is known as Waterbirds (Sagawa et al., 2019). It is a semi-synthetic dataset of images of two types of birds placed on two types of backgrounds. We embed the images with a pre-trained ResNet50 (He et al., 2016) model. To obtain gradient space representations, we trained logistic regression models. See Section 4 for additional details.

We consider three popular clustering methods: K-means, DBSCAN with Euclidean distance, and DBSCAN with centered cosine distance. Group annotations quality is evaluated using the Adjusted Rand Index (ARI) (Hubert & Arabie, 1985), a measure of clustering quality. Higher ARI indicates higher group annotations quality, and ARI = 1 implies the predicted group partition is identical to the true group partition. The definition of ARI is provided in Appendix A. We summarize the results in Table 2. Clustering in the gradient space noticeably outperforms clustering in the feature space. These results provide empirical evidence that gradient space facilitates learning of group annotations via clustering. Visualization of the feature space and gradient space of the Synthetic and Waterbirds datasets are provided in Appendix B.

## 3.2 GRASP FOR GROUP INFERENCE AND OUTLIER IDENTIFICATION

Having motivated our choice of performing clustering in the gradient space, we now present GRASP in detail. We then describe how to train a distributionally and outlier robust model using GRASP.

**Clustering method and distnace measure.** Results in Table 2 indicate that both K-means and DBSCAN perform well in the gradient space. DBSCAN is a density-based clustering algorithm, where clusters are defined as areas of higher density, while the rest of the data is considered outliers. In this work, we choose to use DBSCAN for its ability to identify outliers, which is an important aspect of the problem we consider. As an additional benefit, unlike K-means, it does not require knowledge of the number of groups. See Appendix A for a detailed description of DBSCAN.

In terms of distance measure, we recommend cosine distance due to its better performance on the Waterbirds data, which closer resembles real data. We note that the distance and clustering method choices could be reconsidered depending on the application. For example, for Gaussian-like data without outliers, K-means performed better in Table 2.

**GRASP.** We present the pseudocode of GRASP in Algorithm 1. We first train an ERM classifier $h_{\boldsymbol{\theta}}(\cdot)$ and collect the gradients of sample's loss w.r.t. model parameters $\boldsymbol{\theta}$. We then compute the pairwise centered cosine distances within each class $y \in \mathcal{Y}$ using gradient representations, as discussed in Sec. 3.1. Lastly, to estimate the group annotations and identify outliers, we apply DBSCAN on these distance matrices for each class $y \in \mathcal{Y}$.

**Training models with improved worst-group performance in the presence of outliers using GRASP.** We discard the identified outliers and then provide learned group annotations as inputs to a Group-aware method of choice. For concreteness, in this work, we will use (gDRO). Specifically, we employ the method of Sagawa et al. (2019) to solve the gDRO problem, which is a stochastic

optimization algorithm with convergence guarantees. We note that other choices could be appropriate. For example, methods accounting for noise in group annotations (Lamy et al., 2019; Mozannar et al., 2020; Celis et al., 2021) are interesting to consider as they could counteract mistakes in GRASP annotations.

**Remark.** We note that the model $h_\theta$ and parameter space $\Theta$ used for computing gradient representations $f$ and learning group annotations with GRASP can be different from the classifier and parameter space used for the final model training. For example, one can train a logistic regression model (using features from a pre-trained model when appropriate) and collect the corresponding gradients for GRASP, and then train a deep neural network of choice with the estimated group annotations.

## 4 EXPERIMENTS

In this section, we conduct extensive experiments on both synthetic and benchmark datasets to evaluate the performance of GRASP.[3] Our results show that GRASP outperforms the state-of-the-art baselines in terms of group identification quality and downstream worst-group performance while providing robustness to outliers.

### 4.1 DATASETS AND BASELINES

**Synthetic.** We generate a synthetic dataset of 1,000 samples with two features $\mathbf{x} \in \mathbb{R}^2$, a group attribute $g \in [4]$, and a binary label $y \in \{0, 1\}$, similar to the motivating example of Figure 1. **(Clean):** The synthetic dataset consists of 10 Gaussian clusters with a variance of 0.01, and each Gaussian cluster contains 100 samples. Class 0 is divided into two groups: group 3 consists of four Gaussian clusters with centers $(1, 5), (1, 3), (1, 2), (1, 1)$; group 2 consists of one Gaussian cluster with center $(0, 4)$. Similarly, Class 1 is divided into two groups: group 1 consists of four Gaussian clusters with centers $(0, 5), (0, 3), (0, 2), (0, 1)$; group 2 consists of one Gaussian cluster with center $(1, 4)$. **(Contaminated):** We contaminate the synthetic dataset by flipping randomly selected 5% of labels. The contaminated synthetic dataset is visualized in Appendix Figure 5a.

**Waterbirds. (Clean):** Waterbirds (Sagawa et al., 2019; Wah et al., 2011) is a semi-synthetic image dataset of land birds and water birds (Wah et al., 2011) placed on either land or water backgrounds using images from the Places dataset (Zhou et al., 2017). There are 11,788 images of birds on their typical (majority) and atypical (minority) backgrounds. The task is to predict the types of birds and the background type is the group (2 background types per class, a total of 4 groups). We follow an identical procedure to Idrissi et al. (2022) to pre-process the dataset. **(Contaminated):** We contaminate the Waterbirds dataset by introducing outliers in the training and validation datasets. We flip the class labels of 2% of the data, transform 1% of the images with Gaussian blurring, color dither (randomly change the brightness, saturation, and contrast of the images) 1% of the images, and posterize 1% of the images maintaining 4 bits per color channel. We visualize a contaminated example in Appendix Figure 5c.

**COMPAS & CivilComments.** Both datasets are real and collected by humans, therefore likely to contain outliers. **(COMPAS):** COMPAS (ProPublica, 2021) is a recidivism risk score prediction dataset consisting of 7,214 samples. Each class $y \in [0, 1]$ is divided into six groups: Caucasian males, Caucasian females, African-American males, African-American females, males of other races, and females of other races, making 12 groups in total. **(CivilComments):** CivilComments (Dixon et al., 2018; Koh et al., 2021) is a language dataset containing online forum comments. The task is to predict whether comments are toxic or not. We follow a procedure identical to Idrissi et al. (2022) to preprocess the dataset. We divide comments in each class into two groups according to the presence or absence of identity terms pertaining to protected groups (LGBT, Black, White, Christian, Muslim, other religion).

**Experimental baselines.** We compare GRASP to four different types of baselines: (1) standard empirical risk minimization (ERM), (2) a group-aware method (gDRO (Sagawa et al., 2019)), (3) a group-oblivious method (DORO, CVaR-DORO variation (Zhai et al., 2021)), and (4) two group-learning methods (EIIL (Creager et al., 2021), George (Sohoni et al., 2020)). We chose DORO among

---

[3]Our code is available in Github repository `https://github.com/yzeng58/private_demographics`.

Table 3: **Group identification performance of group-learning methods measured by Adjusted Rand Index (ARI).** Higher ARI indicated higher group identification quality. The results are reported on clean and contaminated versions of Synthetic and Waterbirds datasets, COMPAS and CivilComments datasets. GRASP significantly outperforms the other group-learning baselines on all the tested datasets. Moreover, we observe that GRASP is robust to outliers.

| | SYNTHETIC | | WATERBIRDS | | COMPAS | CIVILCOMMENTS |
| OUTLIERS? | ✗ | ✓ | ✗ | ✓ | | |
| METHOD | ARI | ARI | ARI | ARI | ARI | ARI |
| --- | --- | --- | --- | --- | --- | --- |
| EIIL | -.0069 | -.0043 | .0114 | .0078 | -.0025 | -.0001 |
| GEORGE | .6027 | .4565 | .2832 | .2600 | .1962 | .1422 |
| FEASP | .5133 | .4946 | .0418 | .0418 | .2956 | .2093 |
| GRASP (OURS) | **.6943** | **.6944** | **.7453** | **.7453** | **.5453** | **.2639** |

the methods relying on loss values to improve worst-group performance because it is the only method from this group designed to be robust to outliers. Recall that only the group-aware method (gDRO) has access to the true group annotations, thus it should be interpreted as an "oracle" baseline.

We also perform an ablation study by considering an additional group-learning baseline, Feature Space Partitioning (FeaSP). It is identical to GraSP except it performs DBSCAN clustering in the feature space. Comparison to FeaSP emphasizes the importance of clustering in the gradient space as opposed to other choices such as the clustering method and distance measure.

## 4.2 EVALUATION OF GRASP

In this section, we assess the performance of GRASP in terms of group identification and downstream tasks of training models with comparable performance across groups, both with and without outliers. In all experiments, we consider true group annotations unknown in both train and validation data (except for "oracle" gDRO which has access to true group annotations in both train and validation data). Arguably, this setting is more practical due to group annotations often being expensive or infeasible to obtain, even for a smaller validation set. We note that this setting differs from the majority of prior works considering unknown group annotations (see Table 1). For example, inspecting Table 5 in Appendix B.2 of Zhai et al. (2021), we notice that their DORO is unable to improve upon ERM without access to validation data with true group annotations (see results for non-oracle model selection). We report results with known validation group annotations in Appendix C.3.

**Group annotations quality.** The first experiment examines the quality of group annotations learned with GRASP. To collect the gradients of the data's losses w.r.t. the model parameters, we train a logistic regression model on the Synthetic dataset, a three-layer ReLU neural network with 50 hidden neurons per layer on the COMPAS dataset, and a BERT (Devlin et al., 2018) model on the CivilComments dataset (due to the large number of parameters in BERT, we only consider the last transformer and the subsequent prediction layer when extracting gradients). For the Waterbirds dataset, we first featurize the images using a ResNet50 pre-trained on ImageNet (Deng et al., 2009), and then train a logistic regression. We then use DBSCAN clustering with centered cosine distance. We select DBSCAN hyperparameters using standard clustering metrics that do not require knowledge of the true group annotations, see Appendix C.1.

In Table 3, we compare group identification quality of GRASP (measured with ARI) to three group learning baselines, EIIL, George, and FeaSP, across four datasets. There are two key observations supporting the claims made in this paper: (i) clustering in the gradient space (GRASP) outperforms clustering in the feature space (FeaSP and George), as well as other baselines (EIIL); (ii) GRASP is robust to outliers, i.e. it performs equally well in the presence and absence of outliers. To comment on the low ARI of EIIL, we note that the Invariant Risk criteria EIIL optimizes was designed primarily for invariant learning (i.e., learning environment labels) (Arjovsky et al., 2019; Creager et al., 2021), which may not be suitable for learning group annotations.

**Worst-group performance.** The standard metric when comparing methods for training ML models with comparable performance across groups (evaluated w.r.t. true group annotations) is worst-group accuracy (Sagawa et al., 2019; Koh et al., 2021). For the group-learning methods (GRASP, FeaSP,

Table 4: **Downstream worst-group accuracy and average accuracy on the test data.** The average test accuracy is a re-weighted average of the group-specific accuracies, where the weights are based on the training distribution. The results are reported on clean and contaminated versions of Synthetic and Waterbirds datasets, COMPAS and CivilComments datasets. We observe that GRASP significantly outperforms the group-oblivious (DORO) and other group-learning approaches (EIIL, George, FeaSP) methods on Synthetic, COMPAS, and CivilComments datasets, and performs relatively well on Waterbirds datasets, while being robust to outliers.

| OUTLIERS? | SYNTHETIC | | WATERBIRDS | | COMPAS | CIVILCOMMENTS |
| | ✗ | ✓ | ✗ | ✓ | | |
| METHOD | WORST.(AVG.) | WORST.(AVG.) | WORST.(AVG.) | WORST.(AVG.) | WORST.(AVG.) | WORST.(AVG.) |
|---|---|---|---|---|---|---|
| ERM | .6667(.8823) | .5333(.8273) | .6075(.9673) | .5249(.9621) | .4706(.6792) | .4659(.9213) |
| DORO | .6667(.8823) | .6000(.8342) | .5888(.9694) | .6636(.9686) | .4706(.6801) | .4905(.9182) |
| EIIL | .6667(.8783) | .6000 (.8115) | .6916 (.9645) | **.7056(.9629)** | .0588 (.6046) | .6056(.9066) |
| GEORGE | .5333(.8732) | .6000(.8342) | **.7523**(.9612) | .5897(.9100) | .4416(.6232) | .5897(.9100) |
| FEASP | .6667(.8823) | .6667(.8823) | .1417(.9346) | .1417(.9346) | .4416(.6232) | .6056(.9066) |
| GRASP (OURS) | **.8000**(.8926) | **.8000**(.8926) | .6854(.9654) | .6798(.9004) | **.4743**(.6717) | **.6798(.9004)** |
| GDRO (ORACLE) | .7333(.8639) | .8000(.8755) | .8665(.9272) | .8545(.9081) | .4625(.6807) | .6941(.8767) |

George, EIIL), we first discard identified outliers if applicable (GRASP and FeaSP), and then train gDRO with the corresponding learned group annotations. We also use the learned group annotations on the validation data to select the corresponding gDRO hyperparameters. In Appendix C.2 we demonstrate that GRASP worst-group performance is fairly robust to the corresponding DB-SCAN hyperparameters. For ERM and DORO we used the validation set overall performance for hyperparameter selection.

For all methods, on a given dataset, we train models with the same architecture and initialization. Recall that these models can be different from the models used in estimating group annotations with any of the group-learning methods. See Appendix C.1 for details.

We summarize results in Table 4. GRASP outperforms baselines on Synthetic, COMPAS, and Civil-Comments datasets. For the Waterbirds dataset, GRASP also performs relatively well. Interestingly, EIIL performs best on the contaminated Waterbirds dataset, despite the poor ARI discussed earlier. It is, however, failing on the COMPAS dataset. We also notice that GRASP outperforms "oracle" gDRO on Synthetic and COMPAS datasets. This could be due to the fact that gradient space clustering helps to focus on "harder" instances, as discussed in Section 3.1, while the available ("oracle") group annotations (at least on COMPAS), might be noisy.

## 5 CONCLUSION

In this work, we considered the problem of learning group annotations in the presence of outliers. Our method allows training models with comparable performance across groups to alleviate spurious correlations and accommodate subpopulation shifts when group annotations are not available and need to be estimated from data. We accomplished this by leveraging existing outlier-robust clustering approaches to estimate the group (and outlier) memberships of each point in the dataset. Key to our proposed approach is performing the clustering in the *gradient space*, where the gradient is of the loss at each point with respect to model parameters. We provided strong intuitive and empirical justifications for using the gradient space over the feature space. Finally, we provided a variety of synthetic and real-world experiments where GRASP consistently outperformed or nearly matched the performance of all comparable baselines in terms of both learned group annotations quality and downstream worst-group performance.

One advantage of the gradient space is the simplification of the structure of the correctly classified points (often the majority group), which is also a limitation if identifying *subgroups* within the majority group is of interest. This does not affect the downstream worst-group performance, but may be undesirable from the exploratory data analysis perspective.

As a next step, when training models with GRASP group annotations, it would be interesting to consider alternatives to gDRO that are accounting for noise in group annotations (Lamy et al., 2019; Mozannar et al., 2020; Celis et al., 2021) to counteract GRASP estimation error. Alternatively, one can consider training with group-oblivious methods such as DORO (Zhai et al., 2021) and performing model selection on the validation data with GRASP group annotations.

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

## A  BACKGROUND

In this section, we provide a mind map of our problem setting (see Fig. 3), the details of the Adjusted Rand Index (ARI), and describe the complete algorithm of DBSCAN.

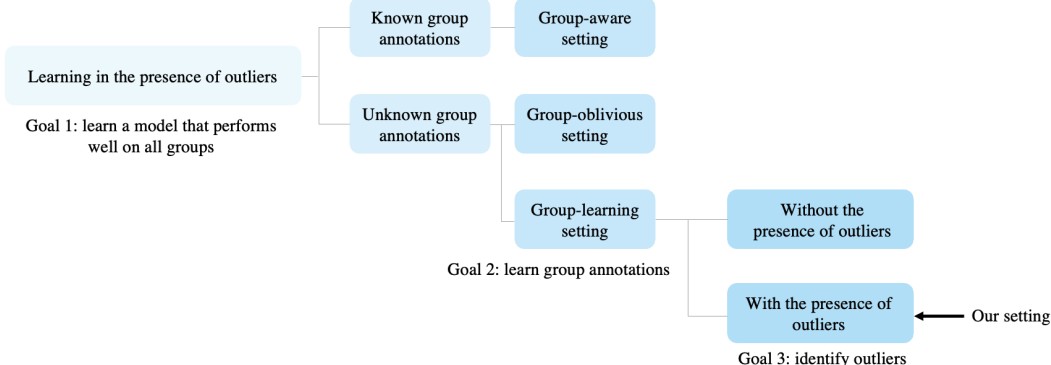

Figure 3: Mind map of our problem setting.

**Adjusted Rand Index (ARI)**   (Hubert & Arabie, 1985) The Adjusted Rand Index (ARI) is a measure of the degree of agreement between two data partitions and accounts for the chance grouping of elements in the data sets. In our case, consider true group partition $P$ and estimated group partition $\hat{P}$. ARI can be computed by

$$ARI(P, \hat{P}) = \frac{\sum_{k,k'} \binom{n_{kk'}}{2} - \left[\sum_k \binom{n_k}{2} \sum'_k \binom{n_{k'}}{2}\right] / \binom{n}{2}}{\frac{1}{2}\left[\sum_k \binom{n_k}{2} + \sum'_k \binom{n_{k'}}{2}\right] - \left[\sum_k \binom{n_k}{2} \sum'_k \binom{n'_k}{2}\right] / \binom{n}{2}},$$

where $n_{kk'}$ is the number of data points belonging to $\mathcal{G}_k \in P$ assigned to group $\hat{\mathcal{G}}_{k'} \in \hat{P}$, $n_k = |\mathcal{G}_k|$, $n_{k'} = |\mathcal{G}_{k'}|$, and $n$ is the total number of samples in the dataset.

**DBSCAN**   (Ester et al., 1996) DBSCAN is a clustering and outlier-detecting method that does not require the number of clusters to be known. It operates on a distance matrix $D$. We call a sample as a "core sample" if there exist $m$ other samples within a distance of $\epsilon$ from this sample. DBSCAN starts with a single cluster that contains an arbitrary core sample and adds core samples from the neighborhood of the cluster to the cluster until all core samples in the $\epsilon$-neighborhood of the cluster have been visited. It then adds the remaining samples in the $\epsilon$-neighborhood of the cluster to the cluster. Next, DBSCAN creates another cluster and expands that cluster by finding unvisited core samples. It then repeats this process of creating and expanding clusters until all core samples have been visited. Any remaining samples that are not added to a cluster are considered outliers. Note that DBSCAN clustering requires two hyperparameters $(\epsilon, m)$ and a distance matrix $D$ as input.

## B  VISUALIZATION OF GRADIENT SPACE AND FEATURE SPACE

In this section, we visualize the gradient space and feature space of contaminated Synthetic (see Fig. 4a) and Waterbirds dataset (see Fig. 4b).

## C  EXPERIMENT

### C.1  MORE DETAILS OF EXPERIMENT SETUP

**Datasets**   The batch size of Synthetic, Waterbirds, COMPAS, and CivilComments datasets are 128, 128, 128, and 32 for both group inference and downstream DRO tasks. We split the Synthetic and

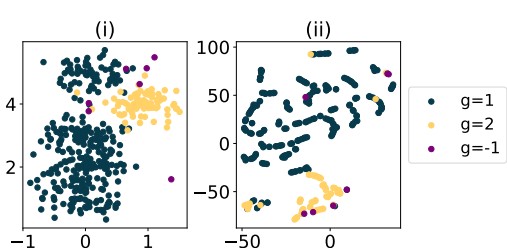

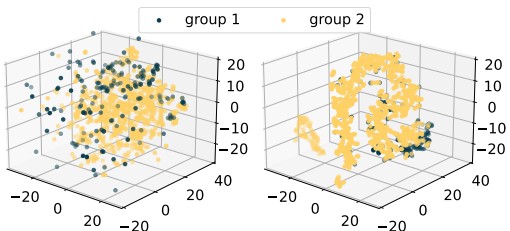

(a) Visualization of input features and 2D t-SNE results of gradients with $y = 0$ on contaminated Synthetic dataset. (i) Input features. (ii) 2D t-SNE results of gradients.

(b) 3D t-SNE visualization of features and gradients with $y = 1$ on contaminated Waterbirds dataset. Left: 3D t-SNE visualization of features extracted from ResNet-50 pretrained on ImageNet (Deng et al., 2009). Right: 3D t-SNE visualization of gradients of the sample's loss w.r.t. the parameters of the last layer.

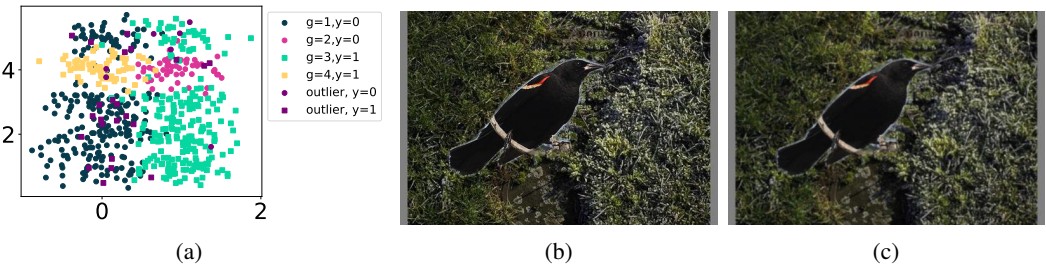

(a)                              (b)                              (c)

Figure 5: (a) Scatter plot of contaminated Synthetic dataset. (b) Original image of `010.Red_winged_Blackbird/Red_Winged_Blackbird_0079_4527.jpg`. (c) Image `010.Red_winged_Blackbird/Red_Winged_Blackbird_0079_4527.jpg` after Gaussian blurring.

COMPAS datasets into training, validation, and test datasets at the ratio of 0.6:0.2:0.2. We follow an identical procedure to Idrissi et al. (2022) to pre-process the Waterbirds and Civilcomments dataset. Fig. 5a visualizes the contaminated Synthetic dataset. We provide an example of a contaminated sample in Fig. 5b and Fig. 5c, which present an image before and after Gaussian blurring.

**Group annotations quality.** To collect the gradients of the corresponding datum's loss w.r.t. the model parameters, we train a logistic regression model on 50 epochs on the Synthetic dataset, a three-layer ReLU neural network with 50 hidden neurons for 300 epochs on the COMPAS dataset, and a BERT (Devlin et al., 2018) model for 10 epochs on the CivilComments dataset. For the Waterbirds dataset, we first featurize the images using a ResNet50 pre-trained on ImageNet (Deng et al., 2009), and then train a logistic regression for 360 epochs We tune the DBSCAN clustering hyperparameters $\epsilon \in \{.1, .2, .3, .5, .7\}, m \in \{10, 20, 30, 50, 70, 100\}$ for each $y \in \mathcal{Y}$, for both FeaSP and GRASP. We tune the learning rate of EIIL in $\{10^{-1}, 10^{-2}, 10^{-3}, 10^{-4}\}$, run EIIL for 50 epochs on Synthetic, Waterbirds, and COMPAS datasets, three epochs on Civilcomments dataset. We tune the overcluster factor of George in $\{1, 2, 5, 10\}$, and employ the over-cluster Gaussian Mixture Model clustering for George. Lastly, we select the best EIIL epoch and other hyperparameters based on *Silhouette Coefficient*, a measure assessing the clustering quality in terms of the degree to which a sample clusters with other similar samples.

**Worst-group performance.** We use Adam optimizer for all trainings. We tune outlier fraction $\epsilon \in \{.005, .01, .02, .1, .2\}$ and minimal group fraction $\in \{.1, .2, .5\}$ for (CvAR-)DORO on all datasets. We tune the learning rate $\in \{10^{-5}, 10^{-4}, 10^{-3}\}$ and weight decay $\in \{10^{-4}, 10^{-3}, 10^{-2}\}$ for all methods. We select the step size of the group weights $q$ in gDRO (Sagawa et al., 2019) $\in \{.001, .01, .1\}$. We train a three-layer ReLU neural network with 50 hidden neurons per layer for 50 and 300 epochs on the Synthetic and COMPAS datasets, respectively. We train a logistic regression model with 360 epochs on the Waterbirds dataset, and a BERT model for 10 epochs on the Civilcomments dataset.

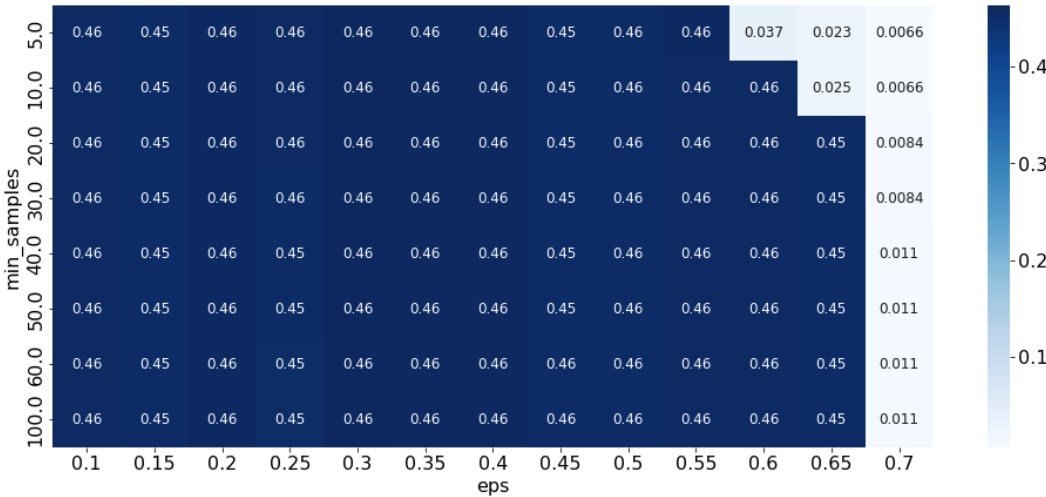

Figure 6: Group identification quality of GRASP v.s. DBSCAN clustering hyperparameters (eps: $\epsilon$, min_samples: $m$) measured in Adjusted Rand Index (ARI) on class 0 of Waterbirds dataset.

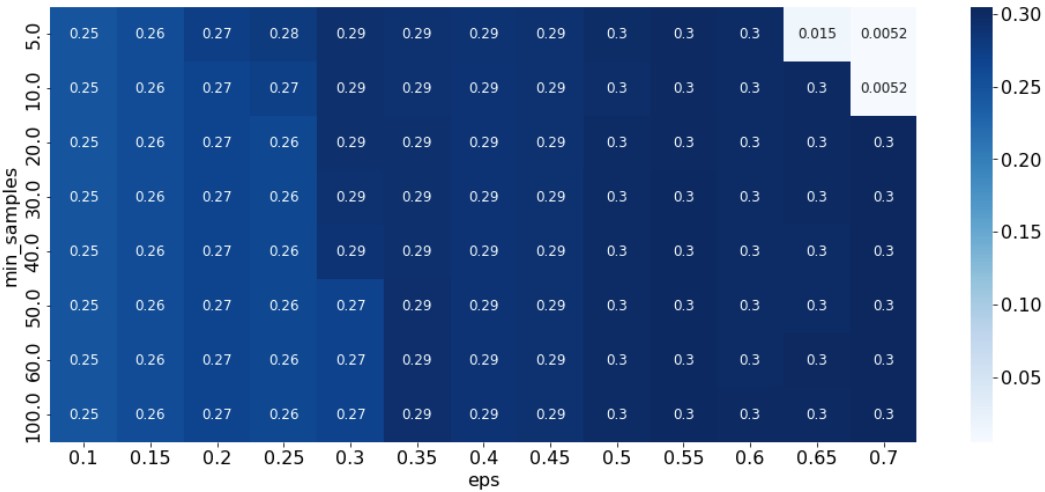

Figure 7: Group identification quality of GRASP v.s. DBSCAN clustering hyperparameters (eps: $\epsilon$, min_samples: $m$) measured in Adjusted Rand Index (ARI) on class 1 of Waterbirds dataset.

## C.2 ROBUSTNESS TO DBSCAN CLUSTERING HYPERPARAMETERS

In this experiment, we investigate the effect of clustering hyperparameters on group inference and downstream DRO task performances. In doing so, we let $\epsilon \in \{.1, .15, .2, .25, .3, .35, .4, .45, .5, .55, .6, .65, .7\}$ and $m \in \{5, 10, 20, 30, 40, 50, 60, 100\}$ and visualize how ARI varies with different choice of clustering hyperparameters on different classes of Waterbirds dataset in Fig. 6 and Fig. 7. We observe that the group identification performance is robust to clustering hyperparameters. For worst-group performance, we set the $\epsilon$ and $m$ to be the same for different classes on the datasets. We visualize how it varies with clustering hyperparameters on Waterbirds and COMPAS dataset in Fig. 8 and Fig. 9. A similar phenomenon is observed for worst-group performance — we find that worst-group performance is fairly robust to DBSCAN clustering hyperparameters.

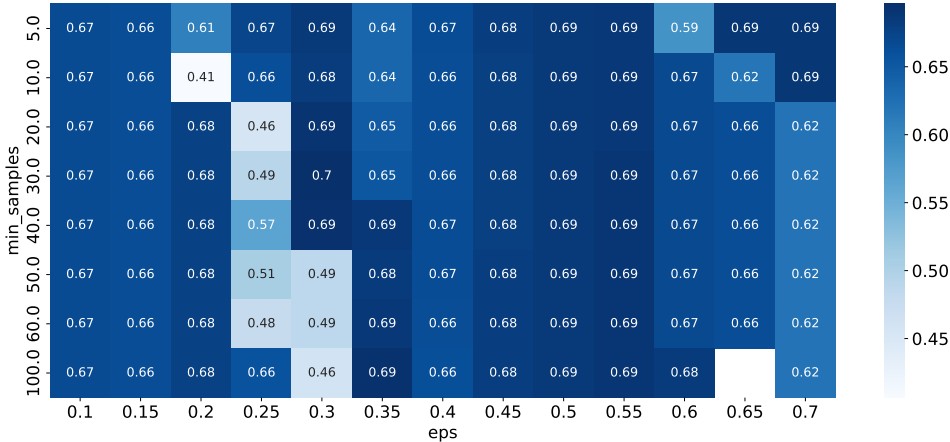

Figure 8: Worst-group accuracy of GRASP v.s. DBSCAN clustering hyperparameters (eps: $\epsilon$,

Figure 9: Worst-group accuracy of GRASP v.s. DBSCAN clustering hyperparameters (eps: $\epsilon$, min_samples: $m$) on COMPAS dataset.

## C.3 RESULTS WITH KNOWN VALIDATION GROUP ANNOTATIONS

Lastly, we report the experiment results on group inference and downstream DRO tasks where the hyperparameters of Group-oblivious (DORO) and Group-learning (EIIL, George, FeaSP, GRASP) are selected based on the corresponding metric computed with true validation group annotations. The same phenomenon we observe under the setting of unavailable validation group annotations also holds under this setting. Table 5 shows that GRASP also significantly outperforms the other Group-learning methods in terms of group learning when true validation group annotations are available. For downstream DRO tasks, while the worst-group accuracy of all Group-oblivious and Group-Learning methods are improved, GRASP still achieves the highest worst-group performance on Synthetic, COMPAS, and Civilcomments datasets and performs relatively well on Waterbirds dataset.

Table 5: **Group identification performance of Group-learning methods measured by Adjusted Rand Index (ARI) when validation group annotations are available.** Higher ARI indicated higher group identification quality. The results are reported on clean and contaminated versions of Synthetic and Waterbirds datasets, COMPAS and Civilcomments datasets. Our GRASP significantly outperforms the other Group-learning baselines on all the tested datasets. Moreover, we observe that GRASP is robust to outliers.

| | SYNTHETIC | | WATERBIRDS | | COMPAS | CIVILCOMMENTS |
| OUTLIERS? | ✗ | ✓ | ✗ | ✓ | | |
| METHOD | ARI | ARI | ARI | ARI | ARI | ARI |
|---|---|---|---|---|---|---|
| EIIL | -.0069 | -.0031 | .0114 | .0078 | -.0025 | -.0001 |
| GEORGE | .6027 | .4565 | .3223 | .3822 | .2059 | .2218 |
| FEASP | .5189 | .5276 | .5189 | .1069 | .2956 | .2072 |
| GRASP (OURS) | **.7497** | **.7241** | **.8137** | **.7531** | **.5453** | **.2863** |

Table 6: **Downstream DRO performance of various methods measured by worst-group accuracy and average accuracy on the test dataset when validation group annotations are available.** The average test accuracy is a re-weighted average of the group-specific accuracies, where the weights are based on the training distribution. The results are reported on clean and contaminated version of Synthetic and Waterbirds datasets, COMPAS, and Civilcomments datasets. We observe that GRASP significantly outperforms the Group-oblivous (DORO) and Group-learning approaches (EIIL, George, FeaSP) methods on Synthetic, COMPAS, and Civilcomments datasets, and performs relatively well on Waterbirds datasets, while being robust to outliers. Note that GRASP sometimes outperforms gDRO (oracle), which can get access to the true group annotations. This is because GRASP may focus on "harder" instances more, which potentially affect the results most.

| | SYNTHETIC | | WATERBIRDS | | COMPAS | CIVILCOMMENTS |
| OUTLIERS? | ✗ | ✓ | ✗ | ✓ | | |
| METHOD | WORST.(AVG.) | WORST.(AVG.) | WORST.(AVG.) | WORST.(AVG.) | WORST.(AVG.) | WORST.(AVG.) |
|---|---|---|---|---|---|---|
| ERM | .6667(.8823) | .5333(.8273) | .6075(.9673) | .5249(.9621) | .4706(.6792) | .4659(.9213) |
| DORO | .6667(.8823) | .7333(.8332) | .6604(.9669) | .6056(.9066) | .4387(.6696) | .6056(.9066) |
| EIIL | .6667(.8783) | .7333 (.8170) | .6927 (.9649) | **.7056(.9629)** | .0588 (.6046) | .6056(.9066) |
| GEORGE | .6667(.8823) | .7333(.8227) | **.8053**(.9511) | .6056(.9066) | .4664(.6219) | .6056(.9056) |
| FEASP | .6667(.8823) | **.8000(.8391)** | .1417(.9346) | .1417(.9346) | .4545(.6386) | .6056(.9066) |
| GRASP (OURS) | **.8000**(.8926) | **.8000**(.8372) | .7274(.9541) | .6804(.8999) | **.4743**(.6681) | **.6804(.8999)** |
| GDRO (ORACLE) | .7333(.8639) | .8000(.8755) | .8665(.9272) | .8545(.9081) | .4625(.6807) | .6941(.8767) |

