# OpenReview forum: "Outlier-Robust Group Inference via Gradient Space Clustering"
_ICLR.cc/2023/Conference — Submitted to ICLR 2023_

### Official Review · Reviewer_tuP3 · 2022-10-22

**Confidence:** 5
**Clarity, Quality, Novelty And Reproducibility:** It may not be so novel.
**Correctness:** 3
**Technical Novelty And Significance:** 2
**Empirical Novelty And Significance:** 2
**Recommendation:** 5

**Strength And Weaknesses:**

Strength: The paper is well written at the beginning and validate features in gradient space is robust to outliers. It seems that numerical is good.
Weakness:
1.They stack the methods of Mirzasoleiman et al., 2020 and Group-learning setting, and then use one classical method DBSCAN to cluster.
2. In comparison of gradient space and feature space, the normalization of the data in Figure 2 is not so clear. I think you do not need to do normalization of the data, since the shrinkage of the correctly classified points is profit to outlier identification.
3. It is not clear what variable the derivative is based on. I thought the gradient is a very large dimensional vector (tensor). Since the network is with many layers (e.g. ResNet-50), the dimension of the derivative should be about 50 times. When moving to numerical result, I found the ResNet-50 is pretrained and the derivative is only related to the parameters of logistic regression.

**Summary Of The Paper:**

The authors address the problem of outlier-robust group inference via gradient space clustering. They extract features first and then calculate the gradient of the extracted features. Finally, they cluster in gradient space.

**Summary Of The Review:**

The authors address the problem of outlier-robust group inference via gradient space clustering. They extract features first and then calculate the gradient of the extracted features. Finally, they cluster in gradient space. But the method just stacks some existing methods together, it is not so novel. I think it is not enough for this top conference.

---

> ### Author Response · Authors · 2022-11-15
> **Response to Reviewer tuP3**
>
> We would like to thank Reviewer tuP3 for your comments as well as the appreciation of our writing and good experiment results.
>
> ---
> > Q1: This work stacks the Crust (`Mirzasoleiman et al., 2020`) and group-learning setting, and then use one classical method DBSCAN to cluster.
>
> Although both our work and Crust (`Mirzasoleiman et al., 2020`) leverage gradients, our work is distinct from Crust in terms of the considered gradients and the hypothesis. Specifically, Crust is motivated by the properties of the Jacobian matrix, i.e., gradients of the model output w.r.t. the model parameters. Unlike the loss gradients considered in our work, Jacobian matrix **is not scaled by the error**. The error term scaling (see eq. 2) in the loss gradient is an essential component motivating our method as discussed in Section 3.1. We provide a more thorough comparison between our approach and Crust in the table below.
>
> | | GraSP (Ours) | Crust (`Mirzasoleiman et al., 2020`) |
> |:---:|:---:|:---:|
> | Goal | Identifying outliers and minority groups | Identifying outliers |
> | Gradient | Gradient of the **classification loss** at each sample with respect to model parameters| Gradient/Jacobian matrix of the **model output** with respect to model parameters |
> | Hypothesis | Gradients within groups behave similarly while outliers exhibit more randomized behavior. | Jacobian spectrum can be split into information space and nuisance space associated with large and small singular values. |
> | Method | Perform clustering in the gradient space | Select data points that provide the best low-rank approximation to the Jacobian matrix |
>
> *References*:
> * Baharan Mirzasoleiman, Kaidi Cao, and Jure Leskovec. Coresets for robust training of deep neural networks against noisy labels. Advances in Neural Information Processing Systems, 33:11465–11477, 2020.
>
> > Q2: Novelty of this work.
>
> We hope the reviewer reconsiders the novelty of clustering in the gradient space in light of the above discussion. To the best of our knowledge, there are no prior works (research papers or evidence of it being used by practitioners) that considered clustering in the gradient (of the loss function w.r.t. the model parameters) space. Further, the problem we are solving is well-motivated by the literature published in top conferences (see our general response) and we propose a state-of-the-art solution that will be of interest to the community.
>
> > Q3: (paraphrased) Compared to Figure 1 which considers original gradient space and feature space with euclidean distance, Figure 2 which considers normalized representations of data with centered cosine distance is not so clear.
>
> We agree that a *naive* application of cosine distance (thus normalization) is not a good idea for clustering in the gradient space. The key is to first center the (unnormalized) gradients as described in **Centered cosine distance** paragraph in Section 3.1. As the equation in the paragraph demonstrates, centering in the gradient space is similar to weighted (by error) centering in the feature space, thus it helps to discount the bias due to the majority group. As a result, the centering is w.r.t. a point that is in between majority and minority groups, thus increasing the angular separation for learning group annotations via cosine distance clustering. The goal of Figure 2 is to illustrate this idea. The effect of centered cosine distance on outlier identification is less apparent and we agree that it is possible to construct examples where outlier detection is hindered by the centered cosine distance. However, our experiments (e.g., Table 2) demonstrate that GraSP with cosine distance performs well especially in higher dimensions (see results for Waterbirds). This is likely due to the structure of high dimensional space, where the image manifolds are intrinsically low dimensional, while the outliers are corrupted in ways that will change the manifold, and generally make their subspace more diverse.
>
> > Q4: It is not clear what variable the derivative is based on.
>
> For scalability and efficiency, we consider a subset of model parameters for large models such as BERT and ResNet-50. As per the reviewer's comment, we clarified this when introducing gradients as data representations in the updated version (see page 4, highlighted in blue).
>
> Here we provide the summary of gradients used in our experiment. For synthetic and COMPAS datasets, we consider the gradient w.r.t. the full model parameters. For scalability and efficiency, for Waterbirds dataset, we use the pretrained ResNet-50 to extract the features, then train a logistic regression, and compute the gradient w.r.t. the parameters of logistic regression; for Civilcomments, we only perform clustering of the gradients w.r.t. the last transformer and the subsequent prediction layer of BERT.

---

> > ### Author Response · Authors · 2022-11-15
> > **Continued: Response to Reviewer tuP3**
> >
> > ---
> > **Final notes**: We hope that our responses address your concerns and ask that you consider increasing your score to support the acceptance our paper. Please let us know if the there are any remaining concerns and we would be happy to discuss. Thanks again for your careful reading!

---

> > > ### Author Response · Authors · 2022-12-08
> > > **Please let us know whether we addressed the respective concerns and questions**
> > >
> > > Dear Reviewer tuP3,
> > >
> > > We thank you for your comments that have helped us to improve the paper. We hope you had a chance to take a look at our responses. We kindly ask if you could please let us know whether we addressed the respective concerns and questions. We look forward to having a fruitful discussion and would appreciate if you consider increasing the score in case our responses have addressed your concerns.

---

### Official Review · Reviewer_zxmE · 2022-10-25

**Confidence:** 4
**Correctness:** 3
**Technical Novelty And Significance:** 3
**Empirical Novelty And Significance:** 3
**Recommendation:** 3

**Clarity, Quality, Novelty And Reproducibility:**

The problem setting of the paper is clear to me. On page 3, the author claim that the goal is to learn the model h. Is identifying the memberships of the groups also the goal of the problem? I am not sure what it means by "group loss" If the authors do not want to provide a detailed formulation, they might want to describe the purpose for such loss.

I am confused by Figure 1. the outlier has the same label as the group g=3. Why is it considered an outlier rather than a member of group g=3? It is also not obvious why the solution in Figure 1(d) better than that in Figure 1(c). Figure 1(c) looks like a more meaningful result than 1(d).

The hypothesis of conducting clusters in the gradient space is unclear to me. I understand that such a method can identify mislabeled data. But I do not think the mislabeled data is equivalent to outliers. Even if the proposed method can identify outliers, a straightforward strategy is to remove these identified outliers and then conduct clustering in the original feature space. I do not understand what the clustering results in the gradient space mean and what the hypothesis behind it is.

On page 8, the authors need to clarify what the worst-group performance means. How are the worst-group accuracy and average accuracy computed?


**Strength And Weaknesses:**

Strengths
The authors proposed to cluster in the space of gradients, which looks novel.

Weaknesses
The problem setting might need to be better explained.
The hypothesis behind clustering in the space of gradients needs to be clarified.
Some details of the experiments need to be elucidated.

**Summary Of The Paper:**

This paper focuses on a clustering problem. The authors proposed to cluster the data in the gradient space. The proposed method is tested on multiple real-world datasets.

**Summary Of The Review:**

Based on the experimental results, the method might be promising. However, I have difficulty in understanding some details of the paper. I believe a better clarification is necessary before the acceptance of this paper.

---

> ### Author Response · Authors · 2022-11-15
> **Response to Reviewer zxmE**
>
> We thank Reviewer zxmE for the detailed reviews. We appreciate that Reviewer zxmE finds our idea novel. Our responses are detailed below.
>
> ----
> > Q1: On page 3, the author claim that the goal is to learn the model h. Is identifying the memberships of the groups also the goal of the problem?
>
> The reviewer is correct that identifying the membership of the groups is also our goal. To further avoid this confusion, here we provide a more clear structure of our problem setting and goals, and the discussion below has been integrated into the updated version.
>
> The goal of our work can be split into three subgoals: (1) learn a model that performs well on all groups, (2) learn group annotations, and (3) identify outliers. Note that if we can predict group annotations (goal 2) and identify outliers (goal 3), we can directly apply group Distributionally Robust Optimization to learn a model that performs well on all groups (goal 1).
>
> ```
> Learning in the presence of outliers
> │ (Goal 1: learn a model that performs well in all groups)
> │
> ├── Known group annotations
> │       │
> │       └── Group-aware setting
> │
> └── Unknown group annotations
>             │
>             ├── Group-oblivious setting
>             │
>             └── Group-learning setting (Goal 2: learn group annotations)
>                     │
>                     ├── Without the presence of outliers
>                     │
>                     └── With the presence of outliers [Our setting]
>                          (Goal 3: identify outliers)
> ```
>
> > Q2: I am not sure what it means by "group loss" if the authors do not want to provide a detailed formulation, they might want to describe the purpose for such loss.
>
> Group loss is defined as the average loss of a group. Formally, for group $k \in [K]$, its group loss is defined as
> $\frac{1}{|G_k|} \sum_{\mathbf{z} \in G_k} \ell (\mathrm{y}, h_{\boldsymbol{\theta}} (\mathbf{x})).$
>  Since one of the goals is to learn a model that performs well on all groups, it is natural to minimize the maximum of group losses: $\max_{k\in [K]}\frac{1}{|G_k|} \sum_{\mathbf{z} \in G_k} \ell (\mathrm{y}, h_{\boldsymbol{\theta}} (\mathbf{x})).$
>
> > Q3: I am confused by Figure 1. The outlier has the same label as the group g=3. Why is it considered an outlier rather than a member of group g=3?
>
> Thanks for pointing this out. The label of the outlier in Figure 1 is $y=0$. We have fixed this typo in the updated version.
>
> > Q4: It is not obvious why the solution in Figure 1(d) better than that in Figure 1(c). I do not understand what the clustering results in the gradient space mean and what the hypothesis behind it is.
>
> Recall that one of our goals is to identify groups correctly. In Figure 1(c), feature space clustering mixes three samples from group $g=1$ and samples from group $g=2$ together, which fails to separate group $g=2$ from other groups and is incorrect. In contrast, Figure 1(d) clusters the samples from the same group together, which recovers the true group partition.
>
> We use Adjusted Rand Index (ARI) which measures the degree of agreement between two data partitions as the metric to evaluate the group identification quality. Higher ARI indicates higher group annotations quality, and ARI = 1 implies the predicted group partition is identical to the true group partition. The ARI of the soluion in Figure 1(c) is 0.308, and the ARI of the solution in Figure 1(d) is 1 > 0.308. Therefore, the solution in Figure 1(d) is better.
>
> > Q5: (paraphrased) The gradient space clustering can identify mislabeled data, but the mislabeled data is not equivalent to outliers.
>
> The gradient space clustering can identify both the data points whose features are far away from the data distribution (e.g., the contaminated Waterbirds dataset) and mislabeled data (e.g., the contaminated Synthetic and Waterbirds datasets) as shown in Table 2. Both types of data can be detrimental for gDRO, thus it is beneficial to identify and remove these data points.
>
> Moreover, mislabeled data can also be considered as outliers (`Fard et al., 2017`). An outlier is defined as a data point that differs significantly from other observations (`Grubbs, 1969`). Clearly, mislabeled data is far away from the main data distribution. We also agree with the reviewer that some literature considers a more narrow definition of outliers, which excludes mislabeled data (`Jadari, 2019`).
>
> *References*:
> * Frank E. Grubbs. Procedures for detecting outlying observations in samples. Technometrics 11.1 (1969): 1-21.
> * Farzaneh S. Fard, Paul Hollensen, Stuart Mcilory, and Thomas Trappenberg. Impact of biased mislabeling on learning with deep networks. In 2017 International Joint Conference on Neural Networks (IJCNN), pp. 2652-2657. IEEE, 2017.
> * Salam Jadari. Finding mislabeled data in datasets: A study on finding mislabeled data in datasetsby studying loss function. (2019).

---

> > ### Author Response · Authors · 2022-11-15
> > **Continued: Response to Reviewer zxmE**
> >
> > > Q6: Why not remove the identified outliers and then conduct the clustering in the original feature space?
> >
> > In fact, even *without* outliers, gradient space clustering works better than feature space clustering. This is supported by experiment results reported in Table 2. Besides, Figure 1 also shows that the main benefit of gradient space clustering compared to feature space clustering is that gradient space clustering can identify minority groups better (see our answer to `Q4`).
> >
> >
> > > Q7: The hypothesis of conducting clustering in the gradient space is unclear.
> >
> > Gradient space simplifies the structure of the majority group, thus it aids in identifying the minority group (see Sec. 3 and Figure 1). We hope that our responses to previous questions helped to elucidate the advantages of the gradient space clustering. Please let us know if you have any further questions.
> >
> > > Q8: On page 8, the authors need to clarify what the worst-group performance means.
> >
> > The worst-group performance refers to the worst performance among the groups, which can be measured by worst-group accuracy. For example, consider a dataset with two groups. The classifier achieves 90\% accuracy on group 1, and achieves 30\% accuracy on group 2. Then the worst-group accuracy of the classifier is 30\%. As per the reviewer's suggestion, we have updated the description of the worst-group performance on page 8, highlighted in blue.
> >
> > ---
> > **Final notes**: We want to thank you again for your comments. We are excited that you find our idea novel and believe that we have addressed all your concerns. In light of our response, we hope that you will consider increasing your score and further support the acceptance of our paper.

---

> > > ### Author Response · Authors · 2022-12-08
> > > **Please let us know whether we addressed the respective concerns and questions**
> > >
> > > Dear Reviewer zxmE,
> > >
> > > We thank you for your comments that have helped us to improve the paper. We hope you had a chance to take a look at our responses. We kindly ask if you could please let us know whether we addressed the respective concerns and questions. We look forward to having a fruitful discussion and would appreciate if you consider increasing the score in case our responses have addressed your concerns.

---

### Official Review · Reviewer_5MtF · 2022-10-26

**Confidence:** 4
**Correctness:** 4
**Technical Novelty And Significance:** 2
**Empirical Novelty And Significance:** 3
**Recommendation:** 5

**Clarity, Quality, Novelty And Reproducibility:**

The organization and clarity of the paper is excellent. The authors build their case carefully, situating the gradient-based clustering strategy with respect to the issues in group annotation learning. The advantages of loss gradient clustering are well-supported with examples and analysis for two common distance measures. The supplementary information gives sufficient information for the experiments to be reproduced.

However, loss gradient clustering cannot be considered as particularly novel. It is a relatively simple (but effective) technique that is likely to be rediscovered by practitioners in many contexts.


**Strength And Weaknesses:**

Strengths:

1) Well-motivated and well-organized. The authors take care to situate the gradient-based clustering strategy with respect to the issues in group annotation learning.

2) A good explanation is given as to why gradient-based clustering can distinguish outlier points as well as minority groups. Such training examples tend to be misclassified, with higher-than-average contributions to the loss, and thereby higher than average gradient magnitudes. The authors analyze the group dispersion effect with respect to Euclidean and centered cosine distance.

3) Loss gradient clustering is shown to be both conceptually simple and practical. The experimental results provide evidence of very significant improvements over clustering in the learned feature space, and is competitive with the state-of-the-art methods for DRO.

Weaknesses:

1) Gradient-based clustering, even for loss functions, is not a new idea. For example, see:

Armacki et al., "Gradient Based Clustering", arXiv 2202.00720, 17 June 2002.

Monath et al., "Gradient-based hierarchical clustering", Discrete Structures in Machine Learning Workshop, NIPS 2017.

2) The proposed group annotation learning is a fairly straightforward application of the loss gradient clustering technique, one which would present little in the way of difficulty to most practitioners.


**Summary Of The Paper:**

This paper is concerned with learning in the presence of relatively small `minority' groups in training data, through the use of clustering of loss gradients.  The authors argue that in addition to its previous uses in identifying outliers in training data, clustering in the gradient space can also be used to identify and annotate minority groups, in preparation for group-aware learning.  As a byproduct, outliers can also be filtered from the training set.  The approach is shown to be competitive with state-of-the-art methods for Distributionally Robust Optimization.


**Summary Of The Review:**

Although the authors have clearly shown the value of loss gradient clustering for group annotation learning, the work may not meet the standard of novelty expected of an ICLR submission.

---

> ### Author Response · Authors · 2022-11-15
> **To Reviewer 5MtF**
>
> We thank Reviewer 5MtF for the feedback. We appreciate your acknowledgment that the paper is well-motivated and well-organized. We also share with the Reviewer the belief that our approach is both conceptually simple and practical.  Please find our answers to your comments and questions below.
>
> ---
>
> > Q1: Gradient-based clustering is not a new idea (e.g., Armacki et al. (2022) and Monath et al. (2017)).
>
> The works of Armacki et al. (2022) and Monath et al. (2017) propose methods to **solve clustering problems using gradient-based optimization**. In our work, we **cluster gradients** with classical clustering techniques to learn group annotations. Despite the similarity in names, the two settings are orthogonal to each other. We provide a more thorough comparison between our gradient space clustering and the clustering via gradient-based optimization methods (Armacki et al. (2022), Monath et al. (2017)) in the table below.
>
> | | Gradient Space Clustering (Our Approach) | Clustering via Gradient-based Optimization (`Armacki et al. (2022)`, `Monath et al. (2017))` |
> |:-----------------:|:-----------------:|:-----------:|
> | Dataset | Samples with labels | Samples without labels  |
> | Clustering Space | **Gradient** Space | Feature Space |
> | Key idea | Perform clustering in the gradient space to learn group annotations for downstream supervised learning task | Use **gradient descent** to optimize (unsupervised) clustering objective function |
> | Considered gradient | Gradient of the classification loss at each sample with respect to model parameters | Gradient of the clustering loss with respect to clustering parameters (e.g., centroids) |
> | Motivation of considering gradient | Easier to distinguish outliers and minority groups| Better scalability and efficiency |
> | Algorithm | (1) Train a ERM model; (2) Compute the gradient of the loss at each sample w.r.t. model parameters; (3) Perform clustering on the gradients  | Perform clustering & use gradient descent to optimize the parameters (e.g., centroids)  |
>
> As one can see, although both of them consider gradients, these gradients and motivations are clearly distinct from each other.
>
> *References:*
> * Aleksandar Armacki, Dragana Bajovic, Dusan Jakovetic, and Soummya Kar. Gradient Based Clustering. arXiv preprint arXiv:2202.00720, 2022.
> * Nicholas Monath, Ari Kobren, Akshay Krishnamurthy, and Andrew McCallum. Gradient-based hierarchical clustering. In 31st Conference on neural information processing systems (NIPS 2017), Long Beach, CA, USA. 2017.
>
> > Q2: Novelty of gradient space clustering.
>
> We hope the reviewer reconsiders the novelty of clustering in the gradient space in light of the above discussion. To the best of our knowledge, there are no prior works (research papers or evidence of it being used by practitioners) that considered clustering in the gradient (of the loss function w.r.t. the model parameters) space. Further, the problem we are solving is well-motivated by the literature published in top conferences (see our general response) and we propose a state-of-the-art solution that will be of interest to the community.
>
>
> ----
> **Final notes**: We are excited that you find our work well-motivated and well-organized. If we have successfully addressed your questions, we would strongly appreciate an increased score. Otherwise, please let us know what experiments and/or revisions we can provide to allay your concerns.

---

### Author Response · Authors · 2022-11-15
**To AC and All Reviewers**

We thank the reviewers for their helpful feedback.

First of all, we appreciate that reviewers found that: (i) the paper is well-motivated (R-5MtF), well-written (R-tuP3), and has excellent clarity and organization (R-5MtF); (ii) the proposed approach is conceptually simple, practical (R-5MtF), and novel (R-zxmE); (iii) the experimental results are good (R-tuP3) and reproducible (R-5MtF).

As for the concerns/questions raised, we believe that we have successfully addressed every single one in the individual responses to the reviewers. **We integrated most of the answers in the newly updated version.** Here we also respond to the comments regarding novelty raised by two of the reviewers below (R-5MtF and R-tuP3).

> Q: (paraphrased) Novelty of the proposed method.

We first compare our work with the previous work on learning in the presence of minority groups as below. All of the prior works listed here address the same problem (some without considering outliers) and have been published in top-tier conferences, and **we propose a state-of-the-art** solution that differs meaningfully from all prior work. We agree with the reviewers that our method is simple. Nevertheless, it is still novel *and* impactful as it solves a well-motivated problem better than multiple prior works.
* `Hashimoto et al. (2018)`: distributionally robust optimization.
* `Sohoni et al. (2020)`: perform feature space clustering and then apply group distributionally robust optimization.
* `Zhai et al. (2021)`: perform distributionally robust optimization, and remove the high-loss samples at each iteration.
* `Liu et al. (2021)`: train a ERM model, and then train the final model by upweighting the samples misclassified by the ERM model.
* `Creager et al. (2021)`: compute the soft group annotations by maximizing the soft per-group risk, and then apply group distributionally robust optimization.
* `Ours`: train a ERM model, perform gradient space clustering, and then apply group distributionally robust optimization.

Moreover, we conduct a more thorough comparison of our work and the related work mentioned by the reviewers (`Mirzasoleiman et al. (2020)`, `Armacki et al. (2022)`, `Monath et al. (2017)`) which raised concerns ragarding the novelty of our work. We show that our work is distinct from theirs in the discussion below.


The works of `Armacki et al. (2022)` and `Monath et al. (2017)` propose methods to solve clustering problems using gradient-based optimization. In our work, we cluster gradients with classical clustering techniques to learn group annotations. The two settings are completely orthogonal to each other, and therefore does not invalidate the novelty of our work. Please see detailed comparison in the table below.

| | Gradient Space Clustering (Ours) | Clustering via Gradient-based Optimization (`Armacki et al. (2022)`, `Monath et al. (2017))` |
|:-----------------:|:-----------------:|:-----------:|
| Dataset | Samples with labels | Samples without labels  |
| Clustering Space | **Gradient** Space | Feature Space |
| Key idea | Perform clustering in the gradient space to learn group annotations for downstream supervised learning task | Use **gradient descent** to optimize (unsupervised) clustering objective function |
| Considered gradient | Gradient of the classification loss at each sample with respect to model parameters | Gradient of the clustering loss with respect to clustering parameters (e.g., centroids) |
| Motivation of considering gradient | Easier to distinguish outliers and minority groups| Better scalability and efficiency |
| Algorithm | (1) Train a ERM model; (2) Compute the gradient of the loss at each sample w.r.t. model parameters; (3) Perform clustering on the gradients  | Perform clustering & use gradient descent to optimize the parameters (e.g., centroids)  |

Although both our work and Crust (`Mirzasoleiman et al., 2020`) leverage gradients, our work is distinct from Crust in terms of the considered gradient and the hypothesis. Specifically, Crust is motivated by the properties of the Jacobian matrix, i.e., gradients of the model outputs w.r.t. the model parameters. Unlike the loss gradients considered in our work, the Jacobian matrix crucially **is not scaled by the error**. The error term scaling (see eq. 2) in the loss gradient is an essential component motivating our method as discussed in Section 3.1. Please see the table below for details.

---

> ### Author Response · Authors · 2022-11-15
> **Continued: To AC and All Reviewers**
>
> | | GraSP (Ours) | Crust (`Mirzasoleiman et al., 2020`) |
> |:---:|:---:|:---:|
> | Goal | Identifying outliers and minority groups | Identifying outliers |
> | Gradient | Gradient of the **classification loss** at each sample with respect to model parameters | Gradient/Jacobian matrix of the **model output** with respect to model parameters |
> | Hypothesis | Gradients within groups behave similarly while outliers exhibit more randomized behavior. | Jacobian spectrum can be split into information space and nuisance space associated with large and small singular values. |
> | Method | Perform clustering in the gradient space | Select data points that provide the best low-rank approximation to the Jacobian matrix |
>
> *References*:
> * Tatsunori Hashimoto, Megha Srivastava, Hongseok Namkoong, and Percy Liang. Fairness without demographics in repeated loss minimization. In International Conference on Machine Learning, pp. 1929–1938. PMLR, 2018.
> * Nimit Sohoni, Jared Dunnmon, Geoffrey Angus, Albert Gu, and Christopher Ré. No subclass left behind: Fine-grained robustness in coarse-grained classification problems. Advances in Neural Information Processing Systems, 33:19339–19352, 2020.
> * Runtian Zhai, Chen Dan, Zico Kolter, and Pradeep Ravikumar. Doro: Distributional and outlier robust optimization. In International Conference on Machine Learning, pp. 12345–12355. PMLR, 2021.
> * Evan Z Liu, Behzad Haghgoo, Annie S Chen, Aditi Raghunathan, Pang Wei Koh, Shiori Sagawa, Percy Liang, and Chelsea Finn. Just train twice: Improving group robustness without training group information. In International Conference on Machine Learning, pp. 6781–6792. PMLR, 2021.
> * Elliot Creager, Jörn-Henrik Jacobsen, and Richard Zemel. Environment inference for invariant learning. In International Conference on Machine Learning, pp. 2189–2200. PMLR, 2021.
> * Aleksandar Armacki, Dragana Bajovic, Dusan Jakovetic, and Soummya Kar. Gradient Based Clustering. arXiv preprint arXiv:2202.00720, 2022.
> * Nicholas Monath, Ari Kobren, Akshay Krishnamurthy, and Andrew McCallum. Gradient-based hierarchical clustering. In 31st Conference on neural information processing systems (NIPS 2017), Long Beach, CA, USA. 2017.
> * Baharan Mirzasoleiman, Kaidi Cao, and Jure Leskovec. Coresets for robust training of deep neural networks against noisy labels. Advances in Neural Information Processing Systems, 33:11465–11477, 2020.

---

> > ### Comment · Reviewer_5MtF · 2022-12-07
> > **Response to authors**
> >
> > I have read the authors' rebuttal, and am responding here rather than under my own review so as to "centralize" the discussion. I appreciate very much the careful situation of the proposed GraSP framework with respect to other recent work involving clustering with gradients, and encourage the authors to integrate this into the main paper. However, the work is still conceptually similar to this work, in particular that of Crust, with the innovation resting on a relatively small technical point (error term scaling). Although the paper certainly has merit, the novelity still seems a bit low for ICLR.

---

> > > ### Author Response · Authors · 2022-12-08
> > > **To Reviewer 5MtF**
> > >
> > > Thank you for starting the discussion! If we understand correctly, the reviewer finds our work not novel enough because we use the high-level idea of "gradient clustering"  that has previously appeared in the Crust paper (`Mirzasoleiman et al., 2020`) in a different problem context. However, we don’t agree that this indicates a lack of novelty in our work or constitutes sufficient ground for rejecting our paper.
> > >
> > > **Firstly, our work has only superficial (largely terminological) similarity to Crust.** Our method is based upon the geometric intuition presented in Figures 1 and 2 specific to our problem setting. As you noted in the review, “The authors build their case carefully, situating the gradient-based clustering strategy with respect to the issues in group annotation learning.” Meanwhile, the ideas motivating Crust and leading them to consider a low-rank Jacobian approximation are orthogonal to ours. Thus, we argue that the similarity in terms of the high-level “gradient clustering” terminology is accidental and superficial as the two methods are motivated by two very different viewpoints on two different problem settings. As a result, the gradient scaling that may appear as a “small technical point,” is indeed a critical distinction motivated by our viewpoint of the problem (as detailed in section 3.1) rather than an incidental small difference. We also note that "gradient clustering", even as a high-level concept, has never appeared in prior works on learning group annotations.
> > >
> > > **Secondly, the reuse of ideas/terminology across problem formulations is behind many influential ML papers and is a recurring theme across science in general.** As a prominent example, adversarial learning, i.e., the generator-discriminator idea, from Generative Adversarial Networks (GANs) (`Goodfellow et al. 2014`) has been adopted in numerous other problem settings, e.g., Domain Adversarial Neural Networks (`Ganin et al., 2016`) for domain adaptation or Adversarial Debiasing (`Zhang et al., 2018`) for algorithmic fairness. In these examples, the model consists of a generator/embedder and a discriminator as in GANs, thus it is similar on a high level, but nonetheless, these works are novel and important to the community in our opinion.
> > >
> > > **Lastly, the simplicity of a method does not necessarily indicate that this method lacks novelty.** We consider the simplicity of our method as a strength as it makes it easier for others to use and build upon it. Prior works give us many examples of influential ideas and methods that were simple and effective, such as Dropout (`Srivastava et al., 2014`), and found numerous use cases, in part due to the ease of using them.
> > >
> > > **To summarize**, our work presents a simple state-of-the-art solution to a well-motivated problem that was previously considered in multiple prior works, including those published at ICML and NeurIPS (`Hashimoto et al., 2018`, `Sohoni et al., 2020`, `Zhai et al., 2021`, `Liu et al., 2021`, `Creager et al., 2021`), and we firmly believe our work will be of interest to the ICLR community.

---

> > > > ### Author Response · Authors · 2022-12-08
> > > > **Continued: To Reviewer 5MtF**
> > > >
> > > > *References:*
> > > >
> > > > * Baharan Mirzasoleiman, Kaidi Cao, and Jure Leskovec. Coresets for robust training of deep neural networks against noisy labels. Advances in Neural Information Processing Systems, 33:11465–11477, 2020.
> > > > * Ian Goodfellow, Jean Pouget-Abadie, Mehdi Mirza, Bing Xu, David Warde-Farley, Sherjil Ozair, Aaron Courville, and Yoshua Bengio. Generative adversarial nets. Advances in Neural Information Processing Systems, 27::2672-2680, 2014.
> > > > * Yaroslav Ganin, Evgeniya Ustinova, Hana Ajakan, Pascal Germain, Hugo Larochelle, François Laviolette, Mario Marchand, and Victor Lempitsky. Domain-adversarial training of neural networks. The journal of machine learning research 17, no. 1: 2096-2030, 2016.
> > > > * Brian Hu Zhang, Blake Lemoine, and Margaret Mitchell. Mitigating unwanted biases with adversarial learning. In Proceedings of the 2018 AAAI/ACM Conference on AI, Ethics, and Society, pp. 335-340. 2018.
> > > > * Nitish Srivastava, Geoffrey Hinton, Alex Krizhevsky, Ilya Sutskever, and Ruslan Salakhutdinov. Dropout: a simple way to prevent neural networks from overfitting. The journal of machine learning research 15, no. 1: 1929-1958, 2014.
> > > > * Tatsunori Hashimoto, Megha Srivastava, Hongseok Namkoong, and Percy Liang. Fairness without demographics in repeated loss minimization. In International Conference on Machine Learning, pp. 1929–1938. PMLR, 2018.
> > > > * Nimit Sohoni, Jared Dunnmon, Geoffrey Angus, Albert Gu, and Christopher Ré. No subclass left behind: Fine-grained robustness in coarse-grained classification problems. Advances in Neural Information Processing Systems, 33:19339–19352, 2020.
> > > > * Runtian Zhai, Chen Dan, Zico Kolter, and Pradeep Ravikumar. Doro: Distributional and outlier robust optimization. In International Conference on Machine Learning, pp. 12345–12355. PMLR, 2021.
> > > > * Evan Z Liu, Behzad Haghgoo, Annie S Chen, Aditi Raghunathan, Pang Wei Koh, Shiori Sagawa, Percy Liang, and Chelsea Finn. Just train twice: Improving group robustness without training group information. In International Conference on Machine Learning, pp. 6781–6792. PMLR, 2021.
> > > > * Elliot Creager, Jörn-Henrik Jacobsen, and Richard Zemel. Environment inference for invariant learning. In International Conference on Machine Learning, pp. 2189–2200. PMLR, 2021.

---

### Decision · Program_Chairs · 2023-01-20

**Decision:**

Reject

**Justification For Why Not Higher Score:**

I have read the paper by myself and also read all the reviews and authors' responses, and carefully considered the assessment of the paper.

I agree with reviewers that the novelty of this paper is not sufficient enough for accepting the paper, although I understand that error term scaling is the original contribution in the proposal as the authors explained in their responses. Therefore this paper is not ready for publication at the moment.

If the authors believe that error term scaling leads to significant effect, deeper theoretical analysis and/or thorough empirical evaluation of the scaling itself would be desired to show the significance of the proposal.


**Justification For Why Not Lower Score:**

N/A

**Metareview: Summary, Strengths And Weaknesses:**

This paper proposes an outlier-robust clustering approach.
The key to the proposal is to perform clustering in not the feature space but the gradient space of model parameters with respect to the classification loss, which can effectively discriminate outliers and minority groups from other data points. Empirical evaluation shows that the proposal is superior to or competitive with other state-of-the-art approaches.

### Strength

The strengths of this paper is three folds:
- The paper is overall clearly written and easy to follow.
- The proposal is well motivated and carefully explained using examples.
- Empirical performance of the proposal seems to be promising.


### Weakness

The major weakness of this paper is the lack of novelty. As reviewers pointed out, the idea of outlier-robust clustering in the gradient space has been already proposed elsewhere, and conceptually similar to Crust by Mirzasoleiman et al., 2020.